

# Effect of high intensity circuit training on muscle mass, muscular strength, and blood parameters in sedentary workers

Sung-Yen Ho[1], Yu-Chun Chung[2], Huey-June Wu[3], Chien-Chang Ho[4,5] and Hung-Ting Chen[1]

[1] Physical Education Office, Ming Chuan University, Taipei, Taiwan
[2] Center for General Education, Taipei Medical University, Taipei, Taiwan
[3] Graduate Institute of Sport Coaching Science, Chinese Culture University, Taipei, Taiwan
[4] Department of Physical Education, Fu Jen Catholic University, New Taipei City, Taiwan
[5] Research and Development Center for Physical Education, Fu Jen Catholic University, New Taipei City, Taiwan

Corresponding author
Hung-Ting Chen,
simonchendr@gmail.com

## ABSTRACT

**Background:** The study aim was to investigate the effect of high intensity circuit training on body composition, muscular performance, and blood parameters in sedentary workers.

**Methods:** A total of 36 middle-aged sedentary female workers were randomly divided into high intensity circuit training (HICT) group, aerobic training (AT) group, and control (CON) group. The exercise training groups performed exercise three times per week for 8 weeks. In HICT, each session was 20–35 min with 2–3 rounds. Rounds were 8 min; the interval between rounds was 4–5 min. In AT, each exercise session comprised 20–35 min of aerobic dance training. Physiological parameters were measured 1 week before and after the interventions. The resulting data were analyzed using two-way mixed design ANOVA, the differences in body composition, muscular performance and blood parameters before and after training were compared.

**Results:** The muscle mass (pre-test: 21.19 ± 2.47 kg; post-test: 21.69 ± 2.46 kg, $p < 0.05$) and knee extension 60°/s (pre-test: 82.10 ± 22.26 Nm/kg; post-test: 83.47 ± 12.83 Nm/kg, $p < 0.05$) of HICT group were significantly improved, with knee extension 60°/s significantly higher than that of the CON group (HICT: 83.47 ± 12.83 Nm/kg; CON: 71.09 ± 26.53 Nm/kg). In the AT group, body weight (BW) decreased significantly (Pre-test: 59.37 ± 8.24 kg; Post-test: 58.94 ± 7.98 kg); no significant change was observed in CON group. The groups exhibited no significant change in blood parameters (hs-CRP, TC, and LDL-C) or IGF-1.

**Conclusions:** Sedentary worker's muscle mass and lower-limb muscular performance were effectively improved by performing 8-week HICT with the benefits of short duration, no spatial constraints, and using one's BW, whereas AT caused a significant decrease in BW. However, the AT induced decrease in BW was probably an effect of muscle loss rather than exercise-induced weight loss.

## INTRODUCTION

A sedentary lifestyle, changes in transportation modes, and the impact of rapid urbanization were the main causes of the increase in physical inactivity (*World Health Organization (WHO), 2021*). In addition, most office workers led a sedentary lifestyle, which was defined as any waking behavior characterized by an energy expenditure ≤ 1.5 metabolic equivalent of tasks (METs) (*World Health Organization (WHO), 2020*). According to *Park et al. (2020)* indicate that sedentary lifestyle has adverse effects on the human body, including increased all-cause mortality, cardiovascular disease, cancer risk, metabolic disorders, osteoporosis and cognitive impairment. If the physically inactive people engaged in physical activity, deaths due to insufficient physical activity could be reduced by an estimated 5.3 million yearly (*Lee et al., 2012*).

To improve the time efficiency of exercise, *Klika & Jordan (2013)* proposed high intensity circuit training (HICT), which combines high intensity interval training (HIIT) and circuit training (CT) to train using body weight (BW). It effectively reduces body fat mass (BF), improves insulin sensitivity, and enhances muscle fitness and maximum oxygen consumption ($VO_2max$) (*Miller et al., 2014*). Another study revealed that the peak oxygen intake ($VO_2peak$), relative $VO_2peak$, heart rate, workload (*Ajjimaporn, Khemtong & Widjaja, 2019*), BMI, waist hip ratio (WHR), skinfold measurement, blood pressure (BP), and mental fatigue assessment of people with a sedentary lifestyle were significantly improved following a 4-week HICT intervention (*Maghade, Diwate & Das, 2019*). According to a study by *Takakura, Masayoshi & Tsubota (2015)*, following 8 weeks of HICT with a frequency of 2–3 times per week, the muscular performance of male and female college students was significantly improved, and the cardiorespiratory endurance of the male students was significantly enhanced. In the past HICT studies, the training duration of each movement was 30 s. However, *Ludin et al. (2015)* conducted 12-week HICT and extended the training duration of each movement to 60 s. The results indicated improvements in $VO_2max$, handgrip strength, and blood glucose but no significant change in BMI, BF, body fat percentage (BFP), or muscle mass. According to these studies, HICT can improve body composition, muscular strength, and cardiorespiratory fitness.

Other studies have conducted HICT through weight training and bike workout has different effects on body composition and blood parameters. *Miller et al. (2014)* conducted a 4-week circuit weight training intervention resulting in significant improvements in systolic blood pressure (SBP), resting heart rate, body fat percentage (BFP), lean body mass, insulin (INS), total cholesterol (TC), and triacylglycerol (TG). In addition, *Paoli et al. (2010)* conducted a 12-week training with a frequency of three sessions per week that comprised circuit high intensity group (CHG), circuit low intensity group (CLG), and endurance group (EG). CHG trained by alternating 8 min of endurance on treadmills (3 min at 65%HRR and 1 min at 75%HRR) with training resistance exercise (underhand cable pulldown, chest press, lateral shoulder raise, horizontal leg press at 6RM with 20 s recovery, and 20 reps abdominal crunch after each resistance exercise perform with three sets). The CLG trained by alternating 8 min of endurance on treadmill at 65%HRmax, and perform the same resistance exercise movements as CHG (three sets of 15RM with 60 s

recovery), and EG trained on treadmills at 65%HR and RPE was maintained between 11 and 13, the duration was 40 min. And the end of running, performed four sets of 20 reps of abdominal crunch. The outcomes indicated that the BW, BFP, WHR, lactate, and upper- and lower-limb muscular performance of the CHG were improved. The same training mode was adopted in another study; significant improvements in BW, BF, diastolic blood pressure (DBP), and parameters such as TC, HDL-C, and LDL-C were observed following a 12-week training (*Paoli et al., 2013*). Accordingly, in addition to the HIIT, various types of HICT are beneficial. Abnormal blood lipids occur when the body is inactive, and the TG, TC, LDL, and glucose in sedentary worker serum are significantly higher than those in non-sedentary workers; thus, they run a higher risk of cardiovascular disease (CVD) (*Ebele et al., 2009*; *Ghosh et al., 2020*).

C-reactive protein (CRP) and insulin-like growth factor 1 (IGF-1) are the blood parameters commonly explored in studies on the influence of exercise interventions on fat and muscle. A crucial factor in inflammation, CRP is secreted mainly by the liver, and elevated the levels are associated with physiological parameters such as BMI (*Guldiken et al., 2007*), waist circumference, and high blood glucose (*González et al., 2006*). A study found that 12 weeks of circuit training (60–80%HRR) had a positive impact on HDL-C, hs-CRP and IGF-1 in elderly obese women with sarcopenia (*Jung et al., 2022*). IGF-1 is generally produced through growth hormone (GH) that activates the PI3K/Akt pathway and stimulates the liver (*Aguirre et al., 2016*); this pathway was proved to induce muscular hypertrophy and reduce the response of regulators for muscle wasting (*Stitt et al., 2004*). Studies have indicated that resistance training not only enhances performance in bench press and leg press but also significantly increase IGF-1 levels (*de Souza Vale et al., 2009*).

In summary, HICT significantly improve metabolic and cardiovascular risk factors such as body fat, BFP, BP, and cholesterol. Moreover, studies of exercise training interventions have indicated that CRP is an indicator of body inflammation, whereas IGF-1 is associated with change in muscle mass. TC and LDL-C reflect blood lipid level. However, sedentary workers have rarely been investigated in studies on HICT, and less discussed are the blood parameters related to body composition and muscular performance. Additionally, past study indicates HICT is highly time-effective, only a limited space and simple equipment (a chair) required. Most resistance training requires professional equipment or free weight training equipment. Our study was designed to compare different exercises training that require only simple equipment, the same duration of training and the similar exercise intensity and RPE. Thus, this study investigated the effect of HICT and AT on body composition, muscular performance, and blood parameters in sedentary workers. We hypothesize that HICT group can improve muscle strength and blood parameters better than AT and CON group.

## MATERIALS AND METHODS

### Experimental approach

A pre-test was conducted in the week following participant recruitment. The test items included body composition (BW, muscle mass, BFP, WHR, and BMI), muscular strength (handgrip strength, back muscle strength, and lower-limb muscle strength), and blood

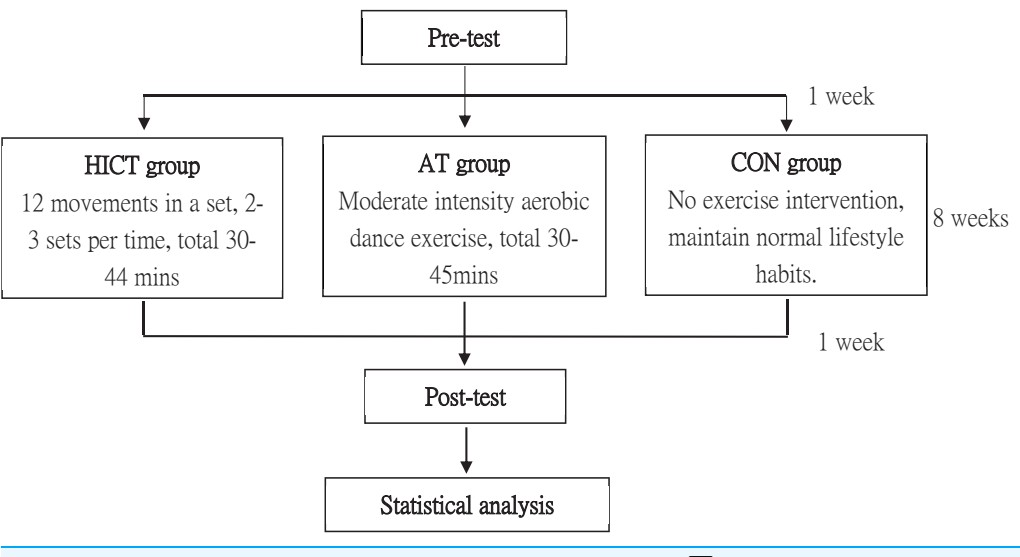

**Figure 1 An overview of the study.**

parameters (hs-CRP, IGF-1, TC, and LDL-C). The participants were then randomly divided into HICT group, AT group, and CON group. The two exercise training groups began an 8-week training program with a frequency of three sessions per week; each session was 20–35 min. The exercise training groups had to complete a total of 24 training sessions. The participants were asked to fast for 12 h prior to blood collection, and their physiological parameters and muscular performance were tested on the next day. A pre-test and a post-test were conducted 1 week before and after the exercise training with the same procedures, items, instruments, and researchers (Fig. 1). To reduce the impact of diet on the result, the subjects must maintain normal diet habit and recorded their diet 3 days before the pre-test and post-test. We received written informed consent from participants of this study. The study was approved by the Fu Jen Catholic University Institutional Review Board (C103069) and carried out in accordance with the Helsinki Declaration.

## Participants

Among the 43 participants recruited, seven failed to complete the study, and the reasons included personal factors preventing complete training, failure to complete the post-test, family factors, and time conflicts. A total of 36 sedentary female workers completed the study (Fig. 2); their average age was 49.97 ± 6.30 years. The inclusion criteria were as follows: (1) 40–64 years old; (2) voluntary participation in the entire research project; and (3) sedentary workers working an average of 8 h a day, 5 days a week, and usually without regular exercise habits. The exclusion criteria were as follows: (1) smoking/alcoholism; (2) suffering from heart or liver disease; (3) major surgery within 1 year; (4) suffering from lower-limb degenerative conditions or injury; (5) suffering from vertigo; (6) BP at rest (SBP/DBP) > 200/110 mmHg; (7) suffering from arterial hypertension; (8) suffering from severe acute illness; and (9) inability to perform moderate/high intensity physical activity.
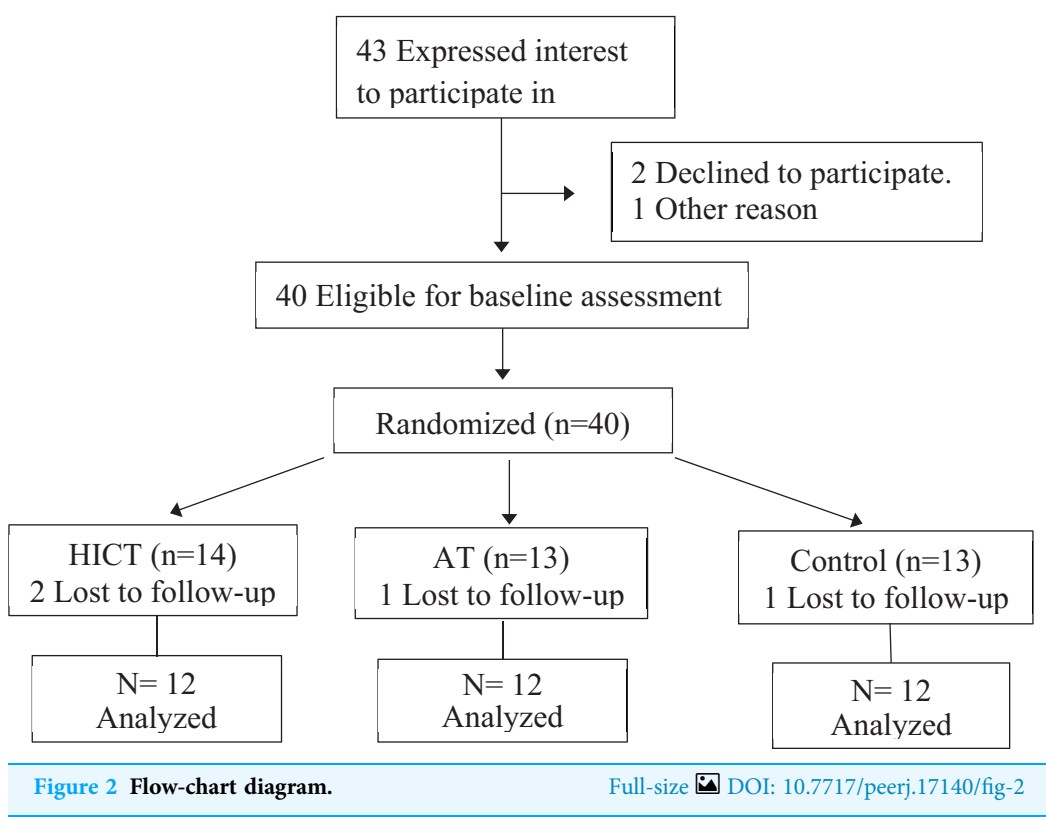

**Figure 2 Flow-chart diagram.**

**Table 1 Format of each training session.**

| Group/Week | Week 1–4 | Week 5–8 |
|---|---|---|
| HICT | HICT: 8 mins × 2 rounds = 16 mins<br>Stretch: 5 min × 2 rounds = 10 mins<br>Rest: 4 mins × 1 = 4 min<br>Total 30 mins | HICT: 8 mins × 3 rounds = 24 mins<br>Stretch: 5 min × 2 rounds = 10 mins<br>Rest: 5 mins × 2 = 10 min<br>Total 44 mins |
| AT | AT: 20 mins<br>Stretch: 5 min × 2 sets = 10 mins<br>Total 30 mins | AT: 35 mins<br>Stretch: 5 min × 2 sets = 10 mins<br>Total 45 mins |

The participants were asked to maintain their original (regular) lifestyle habits and daily routines, meet the test requirements during the study period, and avoid additional exercise.

## Exercise programs

The participants were randomly assigned into three groups. The training groups executed three times per week over 8 weeks, 5-min warmups and 5 min of gentle stretching comprising dynamic and static stretches were performed before and after each training session, respectively. The content of the training was as follows (Table 1).

## HICT group

The training was a variation of the 12 training movements of *Klika & Jordan (2013)*, namely jumping jacks, wall sit, kneeing push-up, abdominal crunch, step-up, squat, triceps

dip on chair, plank, high knee running in place, lunge, T rotation, and left/right side plank. Each movement was completed with the utmost effort for 30 s, with a 10-s interval between movements. From the first to the fourth week, each training session comprised two rounds with a 4-min interval between them, totaling roughly 20 min; from the fifth to the eighth week, each training session comprised three rounds with 5-min intervals between them, totaling approximately 30 min. The number of carotid pulses in 60 s was measured on the right side each round by the subject. The researchers recorded carotid pulses and RPE (rating of perceived exertion).

### AT group

Each AT session of aerobic dance training lasted 20 min from the first to the fourth week and 30 min from the fifth to the eighth week. To control the exercise at a moderate intensity, the number of carotid pulses in 60 s was measured on the right side every 10 min by the subject. The researchers recorded carotid pulses and RPE.

### Control group

The original (regular) lifestyle of the control group was preserved during the 8-week study without any sport training intervention.

### Body composition assessment

A bioelectrical impedance analyzer (Inbody 720; Biospace, Seoul, Korea) was used to measure BW, muscle mass, BFP, BMI, and WHR. To reduce the influence of diet and physical activity, the participants were asked to fast for 12 h prior to the test and to avoid additional physical activity.

### Muscular strength performance

#### Handgrip strength

The participants were in a standing position with both arms hanging naturally at the outer thighs. A digital handgrip dynamometer (DYNAMO METER; TTM, Nagano, Japan) was used to measure the nondominant hand's handgrip strength (kg).

#### Back muscle strength

The participants were asked to stand on a back muscle dynamometer (T.K.K. 5402; BACK-D, Tokyo, Japan) with the body inclined forward approximately 30 degrees. Looking straight ahead with their feet roughly 15 cm apart, they applied lower back force and avoided using instant force or lower-limb force to reduce the risk of falling. The test was conducted twice, and the best result (kg) was recorded.

#### Lower-extremity isokinetic strength

An isokinetic dynamometer (Biodex; Shirley, Boston, MA, USA) was adopted to test the nondominant leg's knee muscular strength (extension/flexion) 60°/s and 180°/s. After warm-up, each test was conducted thrice. The participants were encouraged to exert their utmost effort, and the best relative peak torques (N·M/BW) were recorded.

### Blood collection, storage, and analysis

The participants were asked to fast for at least 12 h. After a quiet rest in a laboratory for 5–10 min, blood in their cubital vein was collected, and the serum obtained following centrifugation was separated with a pipette into Eppendorf tubes and placed in a refrigerator at −80 °C. An enzyme linked immunosorbent assay (ELISA) reader (Epoch; Biotek Inc., Winooski, VT, USA) was used to analyze hs-CRP and IGF-1, and an automated clinical chemistry analyzer (DRI-CHEM 4000i; FUJI, Tokyo, Japan) was used to analyze TC and LDL-C.

### RPE: monitoring exercise intensity

Values of RPE were obtained using the Borg category 6–20 RPE scale (*Nuñez et al., 2020*). The Borg 6–20 scale was used to measure training intensity; six means "no exercise at all," and 20 means "maximal exertion." Following each training session, the participants rated their exertion between 6 and 20 according to level of effort, and the researchers recorded the scores and calculated averages.

## Statistical analyses

The data obtained from the tests were analyzed using SPSS Statistics 20.0 for Windows (Chinese Edition). All the data are presented as mean ± standard deviation (SD). Two-way mixed design ANOVA was employed to compare differences in body composition (BW, muscle mass, BFP, WHR, and BMI), muscular strength performance (handgrip strength, back muscle strength, and lower-extremity strength), and related blood parameters (TC, LDL-C, hs-CRP, and IGF-1) prior to and after the 8-week training. If an interaction was significant, simple main effects analysis was conducted, and the main effects were compared using the least significant difference (LSD) method. The significance level of 0.05 was applied.

## RESULTS

The training completion rates of the HICT and AT groups were 97.2% and 94.8%, respectively. Table 2 provides the demographic data of the groups; no significant differences were noted.

## Body composition

Following 8 weeks of HICT and AT interventions, the HICT group's muscle mass increased significantly ($p = 0.007$), whereas the AT group's BW decreased significantly ($p = 0.045$). However, the groups exhibited no significant change in BFP, WHR, or BMI ($p > 0.05$), and no significant difference in these parameters was observed between the groups ($p > 0.05$) (Table 3).

## Muscular strength performance

Regarding each group's muscular performance after 8 weeks of training, the HICT group's knee extension 60°/s was significantly better than that prior to the training ($p = 0.018$; Hict-pre: 164.80 ± 21.92, Hict-post: 183.85 ± 22.74, AT-pre: 165.02 ± 25.99, AT-post: 170.58 ± 31.23, CON-pre: 156.10 ± 12.38, CON-post: 158.13 ± 14.01), and the HICT group

**Table 2 Participants' descriptive parameters.**

| Items | Groups | (Mean ± SD) | p-value |
|---|---|---|---|
| Age (years) | HICT | 47.67 ± 4.85 | 0.06 |
| | AT | 52.50 ± 5.60 | |
| | CON | 49.09 ± 7.94 | |
| Height (cm) | HICT | 160.50 ± 4.90 | 0.12 |
| | AT | 160.55 ± 4.42 | |
| | CON | 157.36 ± 6.71 | |
| Body weight (kg) | HICT | 58.21 ± 9.60 | 0.22 |
| | AT | 59.37 ± 8.24 | |
| | CON | 58.10 ± 8.79 | |

**Table 3 The participants' body composition before and after training.**

| Items | Groups | Pre-test | Post-test | F value | p-value | |
|---|---|---|---|---|---|---|
| Body weight (kg) | HICT | 58.21 ± 9.60 | 58.57 ± 9.52 | 2.035 | AT = 0.918 | CON = 0.946 |
| | AT | 59.37 ± 8.24 | 58.94 ± 7.98* | 5.136 | HICT = 0.918 | CON = 0.864 |
| | CON | 58.10 ± 8.79 | 58.32 ± 9.11 | 0.504 | HICT = 0.946 | AT = 0.864 |
| Muscle mass (kg) | HICT | 21.19 ± 2.47 | 21.69 ± 2.46* | 7.567 | AT = 0.900 | CON = 0.351 |
| | AT | 21.81 ± 3.07 | 21.55 ± 2.89 | 3.300 | HICT = 0.900 | CON = 0.419 |
| | CON | 20.55 ± 2.91 | 20.63 ± 2.86 | 0.319 | HICT = 0.351 | AT = 0.419 |
| BFP (%) | HICT | 31.58 ± 6.94 | 30.72 ± 6.98 | 2.744 | AT = 0.693 | CON = 0.257 |
| | AT | 31.45 ± 6.83 | 31.77 ± 6.76 | 0.636 | HICT = 0.693 | CON = 0.455 |
| | CON | 33.88 ± 5.51 | 33.76 ±5.55 | 0.076 | HICT = 0.257 | AT = 0.455 |
| WHR | HICT | 0.83 ± 0.05 | 0.84 ± 0.05 | 3.000 | AT = 0.666 | CON = 0.274 |
| | AT | 0.85 ± 0.04 | 0.85 ± 0.03 | 0.071 | HICT = 0.666 | CON = 0.503 |
| | CON | 0.85 ± 0.04 | 0.86 ± 0.04 | 0.314 | HICT = 0.274 | AT = 0.503 |
| BMI (kg/m$^2$) | HICT | 22.62 ± 3.70 | 22.66 ± 3.66 | 0.133 | AT = 0.887 | CON = 0.539 |
| | AT | 23.03 ± 2.53 | 22.84 ± 2.25 | 1.580 | HICT = 0.887 | CON = 0.636 |
| | CON | 23.46 ± 3.18 | 23.45 ± 3.29 | 0.002 | HICT = 0.539 | AT = 0.636 |

**Note:**
* Significantly better than pre-test ($p < 0.05$).

**Table 4 The muscular strength performance before and after training.**

| Items | Groups | Pre-test | Post-test | F value | p-value | |
|---|---|---|---|---|---|---|
| Handgrip strength (kg) | HICT | 25.12 ± 3.37 | 24.69 ± 4.67 | 0.197 | AT = 0.578 | CON = 0.169 |
| | AT | 21.13 ± 5.99 | 23.31 ± 6.83 | 2.278 | HICT = 0.578 | CON = 0.368 |
| | CON | 22.73 ± 5.01 | 21.23 ± 4.67 | 2.497 | HICT = 0.169 | AT = 0.368 |
| Back muscle strength (kg) | HICT | 62.21 ± 11.98 | 64.42 ± 11.14 | 0.380 | AT = 0.072 | CON = 0.055 |
| | AT | 50.13 ± 19.62 | 54.25 ± 15.49 | 0.727 | HICT = 0.072 | CON = 0.898 |
| | CON | 50.67 ± 16.47 | 53.54 ± 13.13 | 0.606 | HICT = 0.055 | AT = 0.898 |

| Table 4 (continued) | | | | | | |
|---|---|---|---|---|---|---|
| Items | Groups | Pre-test | Post-test | F value | p-value | |
| Knee extension 60°/s (Nm/kg) | HICT | 164.80 ± 21.92 | 183.85 ± 22.70*† | 9.678 | AT = 0.193 | CON = 0.019 |
| | AT | 165.02 ± 25.99 | 170.58 ± 31.23 | 1.127 | HICT = 0.193 | CON = 0.217 |
| | CON | 156.10 ± 12.38 | 158.13 ± 14.01 | 0.118 | HICT = 0.019 | AT = 0.217 |
| Knee flexion 60°/s (Nm/kg) | HICT | 93.21 ± 22.80 | 97.80 ± 27.56 | 0.711 | AT = 0.234 | CON = 0.145 |
| | AT | 87.18 ± 17.58 | 92.91 ± 21.96 | 2.790 | HICT = 0.234 | CON = 0.873 |
| | CON | 87.23 ± 20.22 | 91.31 ± 13.31 | 0.564 | HICT = 0.145 | AT = 0.873 |
| Knee extension 180°/s (Nm/kg) | HICT | 82.10 ± 22.26 | 83.47 ± 12.83 | 0.044 | AT = 0.098 | CON = 0.053 |
| | AT | 80.90 ± 28.64 | 82.44 ± 17.71 | 0.064 | HICT = 0.098 | CON = 0.374 |
| | CON | 72.20 ± 27.11 | 71.09 ± 26.53 | 0.015 | HICT = 0.053 | AT = 0.374 |
| Knee flexion 180°/s (Nm/kg) | HICT | 59.84 ± 18.14 | 64.47 ± 16.87 | 0.949 | AT = 0.263 | CON = 0.101 |
| | AT | 54.80 ± 5.64 | 56.98 ± 19.13 | 0.081 | HICT = 0.263 | CON = 0.601 |
| | CON | 52.76 ± 10.33 | 54.29 ± 10.01 | 0.124 | HICT = 0.101 | AT = 0.601 |

Notes:
* Significantly better than pre-test.
† Significantly better than CON after training ($p < 0.05$).

Table 5 The blood parameters before and after training.

| Items | Groups | Pre-test | Post-test | F value | p-value | |
|---|---|---|---|---|---|---|
| TC (mg/dl) | HICT | 157.40 ± 39.92 | 162.20 ± 39.33 | 0.188 | AT = 0.796 | CON = 0.507 |
| | AT | 149.29 ± 50.85 | 156.14 ± 38.81 | 1.081 | HICT = 0.796 | CON = 0.293 |
| | CON | 171.33 ± 47.56 | 176.42 ± 40.00 | 0.176 | HICT = 0.507 | AT = 0.293 |
| LDL-C (mg/dl) | HICT | 114.8 ± 9.23 | 135.5 ± 28.2 | 1.069 | AT = 0.320 | CON = 0.479 |
| | AT | 112.9 ± 23.6 | 127.9 ± 27.4 | 0.518 | HICT = 0.320 | CON = 0.980 |
| | CON | 119.5 ± 11.0 | 138.5 ± 36.4 | 0.323 | HICT = 0.479 | AT = 0.980 |
| hs-CRP (mg/L) | HICT | 0.24 ± 0.41 | 0.25 ± 0.40 | 0.087 | AT = 0.841 | CON = 0.975 |
| | AT | 0.20 ± 0.19 | 0.23 ± 0.20 | 1.133 | HICT = 0.841 | CON = 0.807 |
| | CON | 0.22 ± 0.18 | 0.25 ± 0.26 | 0.504 | HICT = 0.975 | AT = 0.807 |
| IGF-1 (ng/ml) | HICT | 155.63 ± 73.54 | 173.88 ± 83.84 | 2.808 | AT = 0.747 | CON = 0.489 |
| | AT | 170.50 ± 48.19 | 159.00 ± 7.79 | 0.185 | HICT = 0.747 | CON = 0.700 |
| | CON | 157.50 ± 0.71 | 147.75 ± 6.01 | 6.760 | HICT = 0.489 | AT = 0.700 |

Note:
TC, Total cholesterol; LDL-C, low density lipoprotein cholesterol; hs-CRP, High-sensitivity C-reactive protein; IGF-1, Insulin-like growth factor-1.

progressed significantly more than the CON group did. However, no group exhibited significant change in handgrip strength, back muscle strength, knee flexion 60°/s, knee extension 180°/s, or knee flexion 180°/s ($p > 0.05$), with no significant difference between the groups ($p > 0.05$) (Table 4).

## Blood index

Following 8 weeks of HICT or AT, the blood parameters of hs-CRP, TC, and LDL-C as well as IGF-1 exhibited no significant change ($p > 0.05$), with no significant difference between the groups ($p > 0.05$) (Table 5).

## DISCUSSION

After 8 weeks of training, the HICT group's muscle mass and muscular performance in knee extension 60°/s were significantly improved; knee extension 60°/s was significantly better than the CON group. For the AT group, BW decreased significantly. However, the groups exhibited no significant change in other variables of body composition and muscular strength performance, and no significant difference among the groups was observed. In addition, no participant withdrew due to injury, indicating that HICT can be applied safely to sedentary people.

Among prior studies on HICT, in *Klika & Jordan (2013)*, BW was the resistance in HICT performed through 12 movements. Each movement lasted 30 s, and the interval between each movement was 10 s. The entire round was approximately 7 min, and each session comprised 2–3 rounds. The HICT effectively improved muscular fitness. Past studies adopted *Klika & Jordan*'s *(2013)* HICT mode to conduct training. In *Maghade, Diwate & Das (2019)*, the BMI, WHR and BP of obese sedentary workers were significantly improved, while in the study of *Ludin et al. (2015)*, no significant change in the muscle mass, BMI and BF. Moreover, a study by *Paoli et al. (2010)* revealed that, following a 12-week training with three sessions per week, the CHG's BW, BFP, and waist circumference all decreased significantly; the significant decrease in the CHG's body fat was probably due to the higher training intensity leading to oxygen uptake increases after exercise, increasing the time for fat oxidation and improving BFP. The results of another study similarly indicated significant improvement in the HICT group's BW, BF mass, and DBP, and the decreases in the BW of the HICT and low intensity circuit training (LICT) groups were greater than that of the ET group (*Paoli et al., 2013*). *Miller et al. (2014)* conducted a 4-week HICT intervention with an intensity at 8–12 repetition maximum; SBP, BFP, and lean body mass were improved significantly following a training with a frequency of three sessions per week with three rounds in each session. Among these studies, the results obtained by *Ludin et al. (2015)* are similar to those of the present study. However, in comparison to the female college students in *Ludin et al.*'s *(2015)* study, this study investigated middle-aged sedentary workers; the significant increase in their muscle mass following the 8-week training was probably due to their higher muscle atrophy. Regarding the study by *Maghade, Diwate & Das (2019)*, likely because the participants were obese sedentary workers, their BMI and WHR were significantly improved. In addition, the participants of three studies (*Miller et al., 2014*; *Paoli et al., 2010*, *2013*) were obese, and an additional load of resistance was thus added in the HICT mode, amounting to a total load greater than the BW adopted in this study, which was a possible reason for the significant improvement in body composition (body weight, body fat mass, BFP, waist circumference, lean body mass, and BP). In addition, in this study, the AT group's BW significantly decreased, whereas muscle mass and BFP were not improved. In prior studies, BW dropped significantly after the training intervention, along with improvement in BF (*Paoli et al., 2010*, *2013*), whereas in the current study, only the AT group's BW decreased without improvements in muscle mass.

Regarding muscular performance, in this study, only the knee extension 60°/s of the HICT group was significantly improved and notably better than that of the CON group, whereas no significant improvement was noted in the HICT group's handgrip strength, back muscle strength, or knee extension/flexion at the other angular velocity. The significant progress of the HICT group's knee extension 60°/s was probably due to the inclusion of several lower-limb movements in the training, such as wall squat, step-up, half squat, running with elevated legs, and lunge, which strengthened the quadriceps, leading to the significant progress of muscular performance in the knee extension 60°/s. A study used HICT for training (*Takakura, Masayoshi & Tsubota, 2015*) showed that college students' sit-up and push-up performance improved significantly. Compared with studies on traditional resistance training (*Paoli et al., 2010*) such as lat pulldown, crunch, chest press, lateral shoulder raise, horizontal leg press, the muscle strength performance of leg press and cable pulldown improved significantly after training. In the study by *Ludin et al. (2015)*, the participants were overweight or obese female college students, and the training movements were similar to those of this study, but each movement duration was extended from 30 to 60 s. The handgrip showed significant difference than control group after 12-week exercise training. In the present study, the absence of a significant increase in the handgrip strength of the groups was probably due to there was no compensatory movement with fingers supporting BW. No improvement was made in knee flexion, handgrip strength, or back muscle strength, future studies should enhance the training of these parts and consider the inclusion of resistance training equipment such as resistance bands, kettlebells, or dumbbells to enhance training intensity and effectiveness.

No significant change was observed in the blood parameters TC, LDL-C, and hs-CRP or in IGF-1 after 8 weeks of HICT and AT interventions. *Ludin et al. (2015)* adopted training movements like those in this study to conduct training; the results indicated no significant change in TG or TC level. *Miller et al. (2014)* indicated larger improvement in blood cholesterol (TC and TG) when the participant was overweight or obese after the 4 weeks of training, the exercise intervention involved an additional load of resistance, and the total duration of exercise was longer. In the study by *Paoli et al. (2013)*, after a 12-week training program, blood parameters including TC, TG, and LDL were all significantly improved, probably due to longer exercise duration, the additional load of resistance, and the overweight or obesity of the participants, whereas the BMI of the participants in this study were in the reference range and, consequently, no improvement was observed in TC or LDL-C. Moreover, among prior studies on CRP levels in the blood, those of *Nalcakan (2014)* and *Schjerve et al. (2008)* adopted HIIT and reported no change in blood CRP level; the result of the present study is similar. By contrast, *Kamal & Ragy (2012)* conducted interventions of 12-week moderate-intensity aerobic exercise, and CRP levels decreased significantly; the improvement in CPR levels probably occurred because the participants in the studies were obese or overweight, and they adopted moderate-intensity aerobic exercise with longer durations.

Prior research on the influence of exercise intervention on IGF-1 levels, the outcomes obtained by *Jung et al. (2022)* indicated that IGF-1 levels rose significantly following a 12-week circuit training (60–80%HRR) in elderly obese women. However, another study

(*Schiffer et al., 2009*) reported that IGF-1 levels decreased significantly following 12 weeks of moderate-intensity endurance training (lactate threshold at 80% heart rate) or high-load resistance training (70–80%1RM). According to the study, the fatigue induced by high intensity resistance training was a possible reason for lowered IGF-1 levels after training. The two exercise training groups in the present study received only training with BW as the resistance load, and no significant increase was found in IGF-1 levels.

## CONCLUSIONS

In this study, sedentary workers performed 8-week of HICT, and their muscle mass and knee extension 60°/s were significantly improved. In the AT group, BW decreased significantly. The characteristics of the HICT are short duration, using BW as resistance, simple equipment, and no limited by space, and can significantly improve the muscle mass and knee extension performance in sedentary workers. Future studies are suggested to include additional resistance in the training mode for more effective improvements in muscle mass, BF, and blood parameters regarding body composition.

## ACKNOWLEDGEMENTS

We thank Zhi-Yu Wang and Yu-Chun Liu for the recruitment of the participants. We also thank all the volunteered for this study.

### Funding

The authors received no funding for this work.

### Competing Interests

The authors declare that they have no competing interests.

### Author Contributions

- Sung-Yen Ho conceived and designed the experiments, performed the experiments, prepared figures and/or tables, and approved the final draft.
- Yu-Chun Chung conceived and designed the experiments, performed the experiments, analyzed the data, prepared figures and/or tables, and approved the final draft.
- Huey-June Wu performed the experiments, analyzed the data, authored or reviewed drafts of the article, and approved the final draft.
- Chien-Chang Ho analyzed the data, authored or reviewed drafts of the article, and approved the final draft.
- Hung-Ting Chen conceived and designed the experiments, performed the experiments, authored or reviewed drafts of the article, and approved the final draft.

### Human Ethics

The following information was supplied relating to ethical approvals (*i.e.*, approving body and any reference numbers):

Fu Jen Catholic University Institutional Review Board.

## Data Availability

The raw measurements are available in the Supplemental File.

## Supplemental Information

Supplemental information for this article can be found online at http://dx.doi.org/10.7717/peerj.17140#supplemental-information.

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
