# Peer review of "Effect of high intensity circuit training on muscle mass, muscular strength, and blood parameters in sedentary workers"

_PeerJ, doi:10.7717/peerj.17140_

## Round 0.1 · original submission · Major Revisions

Dear Authors,
Following a comprehensive review by two expert referees, it has been determined that your manuscript requires significant revisions. I encourage you to carefully consider and incorporate all the comments provided by the referees in your resubmission.

Thank you for your attention to this matter, and I look forward to receiving your revised manuscript.

Reviewer 1 ·

Basic reporting

Authors conducted an interesting study comparing HICT vs. AT on carious outcomes. I commend to authors on conducted such a large scale training study. Some comments are provide below which I hope prove helpful.

Experimental design

1. Methods Lines 147-149: Can authors clarify what is meant by utmost effort? Also, if the effort wasn’t given were they made to repeat the exercise? This is what I think is meant by the “or repeated for 30 seconds? How was utmost effort assessed?“Each movement was completed with the utmost effort or repeated for 30 seconds, with a 10-second interval between movements.”

2. Methods: Is a set considered one rotation through all of the exercises included? “Rounds” may be more appropriate then “sets”.

Validity of the findings

1. Results, Strength: It may be nice to report the mean difference between groups here. “Regarding each groups muscular performance after 8 weeks of training, the HICT groups knee extension 60o/s was significantly better than that prior to the training (p = .018), and the HICT group progressed significantly more than the CON group did.”

Additional comments

1. Introduction lines 57-58: Please add a citation for this sentence: “It effectively reduces body fat mass (BF), improves insulin sensitivity, and enhances muscle fitness and maximum oxygen consumption (VO2max).”

2. Introduction Lines 66-69: I would add more detail so this study makes sense. It sounds like they were only performing 30s or 60s of exercise when in fact they were performin a circuit with 30s or 60s per exercise. “Ludin et al. (2015) conducted 12-week HICT and extended the training duration from 30 to 60 seconds; the results indicated improvements in VO2max,handgrip strength, and blood glucose but no significant change in BMI, BF, body fat percentage (BFP), or muscle mass.

3. Introduction Lines 74-77: I would add a little description to better understand what authors conducted in this study. Was it high intensity and low intensity resistance training? Was it high intensity and low intensity bodyweight circuit training? Was it high intensity and low intensity something else? It is not clear in the intro if circuit always implies some kind of bodyweight training or could be something different.“In addition, Paoli et al (2010) conducted a 12-week training with a frequency of three sessions per week that comprised circuit high intensity group (CHG), circuit low intensity group (CLG), and endurance group (EG).”

4. Introduction Lines 93-99: I feel like the end of the introduction loses focus a little. Authors began speaking about aerobic and resistance exercise outside of the context of circuit training and I am not sure how it connects to the purpose of the present manuscript. “Aerobic and resistance exercises significantly lower CRP levels (Stewart et al., 2007). IGF-1 is generally produced through growth hormone (GH) that activates the PI3K/Akt pathway and stimulates the liver (Aguirre et al., 2016); this pathway was proved to induce muscular hypertrophy and reduce the response of regulators for muscle wasting (Stitt et al., 2004). Studies have indicated that…”

5. Discussion: Overall the discussion feels a little unorganized. Many studies are discussed and sometimes it is confusing if the authors are talking about their findings or the findings of another study. It might be better to cut down some of the discussion a little bit.

6. Discussion: A loss of bodyfat in the HICT group was not motioned in the results. “The HICT effectively reduced body fat and improved muscular fitness.”

7. Discussion lines 250-251: This sentence does not make sense as written. “Using the training movements of Klika and Jordan (2013) and Ludin et al. (2015) conducted a 12-week training with three sessions per week.

8. Discussion lines 295-298: I don’t think that grip strength really increases even when engaging in resistance training. (see this paper: Rhodes E, Martin A, Taunton J, et al. Effects of one year of resistance training on the relation between muscular strength and bone density in elderly women. Br J Sports Med 2000;34:18–22. Crossref, Medline) Maybe authors should not expect it to increase as this may be uncommon?

9. Discussion: How does the change in strength observed in the HICT group compare to other studies? How does it compare to more traditional resistance training?

10. Discussion lines 323-324: Did muscle mass increase or lean body mass increase? I think the device used measures lean body mass, which could possible be water and other “lean” components. “In the current study, only the HICT group.s muscle mass increased significantly…”

11. Discussion: Were acute changes in IGF-1 measured? I feel like there is too much discussion on IGF-1. Authors should more so focus on which type of exercise can lead to muscle growth. Many loading schemes can induce growth so long as there is sufficient fatigue. Here are a few papers:

Ozaki, H., Loenneke, J. P., Buckner, S. L., & Abe, T. (2016). Muscle growth across a variety of exercise modalities and intensities: Contributions of mechanical and metabolic stimuli. Medical hypotheses, 88, 22-26.

Roberts, M. D., McCarthy, J. J., Hornberger, T. A., Phillips, S. M., Mackey, A. L., Nader, G. A., ... & Esser, K. A. (2023). Mechanisms of mechanical overload-induced skeletal muscle hypertrophy: current understanding and future directions. Physiological reviews, 103(4), 2679-2757.

Reviewer 2 ·

Basic reporting

The current study aimed to investigate the effect of high intensity circuit training (HICT) on body composition, muscular performance, and blood parameters in sedentary workers. Based on the results, HICT protocol promoted muscle hypertrophy and greater strength levels considering knee extension 60o/s, in comparison to sedentary control counterparts. Additionally, aerobic exercise training (AT) reduced body weight, and no adaptive alteration was observed in relation to blood parameters in response to HICT or AT interventions. The current issue is interesting, but multiple points are addressed to the Authors to clarify information and to improve text.
Considering Abstract, in general, this section is well written. Some few points need to be improved; the Results part (lines 29-32) should be accompanied by numerical data to highlight the main findings.
Next, the Introduction section could be improved in terms of formal presentation. In the current form, the background is cumbersome and speculative. The use of HICT and AT (dancing) in the same design is very poorly justified, and the background must be improved to explain the insertion of AT group as a comparison reference. Why was resistance exercise training not considered in the experimental design? Is the dancing exercise a conventional training program previously existent and easier to a controlled conduction?
Regarding references used to support the Introduction, only five (20%) constitute recent articles, published in last five years (from 2018; indeed, this point must be reviewed not only in the Introduction but also in all the manuscript). Consequently, the introductory section must be reviewed and restructured in terms of a logical and direct form, and a primary hypothesis should be described at the end of the Introduction.

Experimental design

Relative to the Materials & Methods, the current study is designed as a clinical trial. Based on this, the clinical trial registry number should be informed. Is it previously registered?
In exclusion criteria, were only hypertensive participants presenting >200/110 mmHg excluded (line 133)? Were other arterial hypertension cases supporting values <200/110 mmHg included into the sample?
What was the randomization method used to allocate the participants into each experimental group? Was there blinding proceeding?
Table 1 should be inserted within Results, and respective p-values from ANOVA in the last column.
Considering exercise programs, training sequences could be better detailed into an illustration, as figure or table; see Skidmore et al. (2012) as a potential example. Moreover, references supporting both exercise training interventions must be included in the manuscript. Relative to warm up and stretching exercise, were they similar in both HICT and AT programs?
With respect to carotid pulses monitoring, post- (or accompanying) exercise palpation of pulse rate may be underestimating exercise heart rate, configuring a potential bias to the measurements. Furthermore, how many researcher(s) made this evaluation? Was an additional method considered to analyze the heart rate?
At least a reference is necessary to support the rating of perceived exertion (RPE) using, as well as all the other methodologies used in this study.

References
Skidmore, B.L.; Jones, M.T.; Blegen, M.; Matthews, T.D. Acute effects of three different circuit weight training protocols on blood lactate, heart rate, and rating of perceived exertion in recreationally active women. J. Sport. Sci. Med. 2012, 11, 660–668.

Validity of the findings

Considering the Results, I’d like to know more about the muscle mass results, more precisely in relation to Two-Way ANOVA and intergroups comparisons. It is noteworthy that HICT presented lower muscle mass than AT group in the Pre-test moment; however, this situation was totally different in the Post-test. In this moment, HICT exhibited higher muscle mass than AT. Since CON and HICT revealed numerically superior results from Pre- to Post-test period, it is very probable that this analysis contains a statistically significant interaction between group and moment, and this was poorly explored into the study. Therefore, Table 2 must be reviewed, and additional symbols (#, † or ‡) should be added to the form.
Likewise, muscle strength results and Table 3 must also be reviewed and reformulated.
Relative to Table 4’s results, authors should research possible intergroups differences based on comparisons among percentage changes from Pre- to Post-test moment. In this case, One-Way ANOVA could be useful to investigate potential exercise training effects. Moreover, were dietary habits monitored?
In relation to Discussion and Conclusions, based on the previous appointments, an important review should be done to improve the current manuscript and show its innovative contribution.

Additional comments

No additional coment.

---

## Round 0.2 · Minor Revisions

Thank you for your resubmission with corrections. While most of the comments have been satisfactorily addressed, there are still a few remaining points that the reviewer has suggested, that need attention. Kindly address these changes and resubmit the revised version.

Reviewer 2 ·

Basic reporting

Based on a major reviewing process, the pointed issues were partially addressed. In particular, the Introduction section was adjusted but it lacks a logical and direct form in terms of focus. This section could be improved. According to the Authors in the Rebuttal Letter: “In the past study indicates HICT is highly time-effective, only a limited space and simple equipment (a chair) required. Most resistance training requires professional equipment or free weight training equipment. This study was designed to compare different exercises training that require only simple equipment, the same duration of training and the similar exercise intensity and RPE”. Certainly, the current text could be inserted within Introduction in order to improve the background.
Other appointments were satisfactorily answered.

Experimental design

No comment.

Validity of the findings

No comment.

Additional comments

No comment.

---

## Round 0.3 · accepted · Accept

Congratulations! You have satisfactorily addressed all reviewer comments.

In the Abstract the authors need to include the units for the outcome measures of muscle mass, body weight/mass and the isokinetic dynamometer.